# In-Person, Remote, or Hybrid Instruction? A Quality Improvement Assessment of a Six Week Interprofessional Education Pathway Program for Undergraduate Pre-Health Students

**DOI:** 10.3390/healthcare10122399

**Published:** 2022-11-29

**Authors:** Asia McCleary-Gaddy, Erica Tengyuan Yu, Robert D. Spears

**Affiliations:** 1McGovern Medical School, University of Texas Health Science Center, Houston, TX 77030, USA; 2Cizik School of Nursing, University of Texas Health Science Center, Houston, TX 77030, USA; 3School of Dentistry, University of Texas Health Science Center, Houston, TX 77030, USA

**Keywords:** pathway programs, interprofessional education, hybrid, diversity, underrepresented in medicine, pre-health, quality improvement

## Abstract

There is an emphasis on increasing the diversity of healthcare providers with the goal of reducing health disparities among racial/ethnic minorities. To support this initiative, pathway programs were designed to provide educational and career support to students belonging to racial/ethnic minorities or those who have challenges applying to or entering health professions. As a consequence of the COVID-19 pandemic, pathway programs have assumed various instructional delivery formats (e.g., face to face, virtual, hybrid) with little knowledge on the satisfaction of such methods. The current preliminary study examines whether in person, virtual, or hybrid learning is most effective for underrepresented pre-health undergraduate students who are engaged in a six-week interprofessional health pathway program. Quantitative and qualitative data was collected at one time point towards the end of the program when it was offered in person, virtually, and in hybrid format. Results revealed that the pre-health pathway program received highest satisfaction when presented in a hybrid format and least satisfaction when presented in virtual instruction. Qualitative data suggests that virtual instruction increases feelings of isolation and complicates educational information due to the limitations of virtual streaming. Implications for pathway design are discussed.

## 1. Introduction

“The pandemic calls for a paradigm shift in our thinking–one that focuses us on delivering health care to all the communities we serve, including the most vulnerable populations”.

David Acosta, MD, Chief Diversity and Inclusion Officer at the Association of American Medical Colleges (AAMC) ABMS Conference 2020-virtual.

As the U.S. healthcare system undergoes transformation, there is an emphasis on promoting diversity among healthcare professionals with the goal of reducing health disparities among racial/ethnic minorities [1]. In response to the call to diversify the healthcare workforce, many academic healthcare institutions have developed pathway programs (formerly known as pipeline programs) to enhance the interest and academic success of underrepresented in medicine (URiM) students for health professions [2,3]. Pathway or pipeline programs are defined as educational and career support to students belonging to racial/ethnic minorities or who have other challenges applying to or entering health professions programs [2,3]. Historically, lack of financial resources, sole focus on standardized test scores vs. holistic admission, lack of URiM identified faculty, and lack of URiM role models have been cited as barriers to racial/ethnic minorities entering health careers. Therefore, pathway programs were created to mitigate these barriers and increase the number of racial/ethnic minority students who apply to a health career [4]. One type of pathway program that has gained popularity over the past few decades due to the collaborative nature of the healthcare environment is interprofessional education (IPE) pathway programs. IPE pathway programs provide students with the opportunity to learn about, from, and with multiple healthcare profession in order to develop interprofessional collaborative working relationships that enable the highest quality of care across settings and therefore optimal health outcomes [2,3].

A prominent theory toward the design of pathway programs for underrepresented students is the Asset Bundle Model. This model suggests that to decrease UriM attrition in the health science pipeline, institutions must address social cues that signal identity devaluation and further develop “asset bundles”. Asset bundles are defined as the specific sets of abilities and resources individuals need to succeed in educational and professional tasks [5]. Collectively, there are five asset bundles; educational endowments, science socialization, network development, family expectations, and material resources [5]. While asset bundles may affect the educational achievement of students from any background, researchers Johnson and Bozeman purport that understanding the interaction of these variables for minorities belonging to multiple stigmatized identity groups is central to the advancement of diversity in healthcare. Specifically, they argue that UriM students who have intersectional identities are more likely to face multilayered challenges as they progress through academic institutions which historically have been environments of systemic oppression [5]. Academic institutions may implicitly convey negative social cues to minoritized students. As a result, attrition out of the pathway toward a healthcare profession may be more likely for students with multiple marginalized social identities [5].

For the purpose of this study, we examined student satisfaction with (1) science socialization-the connection between scientific careers and the ability to serve community goals, (2) network development-building positive social capital through mentoring relationships and involvement in extracurricular activities, students, and (3) educational endowments-access to additional resources and educational support including additional study material and innovative lectures. Science socialization is central to the development of an IPE pathway program because, “if students cannot envision themselves as scientists or health care providers, they will self-select into other paths that seem more viable to them, especially paths that are more consistent with their peers’ and family members’ choices” [5]. Therefore, it is imperative that IPE pathway programs are designed to develop a scientific identity where UriM students are immersed in the norms, behaviors, and social skills applicable to leaders in science and healthcare. This will assist in increasing the self-efficacy of the students and simultaneously demonstrate that their ambitions for a healthcare career are tangible. Moreover, research has shown that relationships with peers, faculty, and staff who are already in healthcare has a positive influence on career outcomes for underrepresented minorities because extensive networks by logic provide greater chances for opportunities. In a study that examined a cascading mentorship model for a medical pathway program, researchers found that by having faculty members from underrepresented backgrounds and mentors who were close to the age of the students, the institution was able to offset potential deficits in the “asset bundles” of science socialization and network development. Using UriM faculty and younger mentors decreased participant concerns of low expectations about academic ability, fear of antagonism from the dominant group, and increased visibility of others with similar backgrounds [6] The current study will expand pastresearch by first examining an IPE pathway program with underrepresented college students but also measuring how mode of instruction affects their satisfaction with the asset bundles.

Although pathway programs have been in existence for decades, in mid-March of 2020, the COVID-19 pandemic had prompted an urgent shift. In one study, 42 of 106 respondents (39.6%) reported canceling some or all of their programs because of the COVID-19 pandemic [7]. During 2020 and 2021, the United States experienced a drastic increase in cases, and governmental leaders took aggressive measures to limit its spread through social distancing-a public health practice that aims to prevent sick people from coming in close contact with healthy people in order to reduce opportunities for disease transmission [8]. As a result of the Centers for Disease Control (CDC) and Prevention’s recommendations to cancel conferences and limit regular meeting sizes, the face-to-face model of educational engagement had been transferred to a remote modality [8]. Remote instruction refers to any educational model in which students complete the program virtually.

It has been reported that immunity can limit the breakout and spread of infection in the population [9]. Specifically, ‘herd immunity’, is the indirect protection from an infectious disease that happens when a population is immune either through vaccination or immunity developed through previous infection. As a result, the increase in rate of COVID-19 vaccinations, academic health science institutions considered a hybrid learning modality. Hybrid learning, is an approach to education that combines online educational materials with traditional in-person classroom methods.

It is difficult to differentiate the efficacy of instruction delivery formats. Emerging research shows that there are various advantages among delivery formats [10]. Face to face instruction (also known as “in person”) can provide deeper understanding through teacher and other students’ body language and voice, while virtual instruction encourages student self-directed learning and to take on more responsibility for their own acquisition of knowledge, while hybrid learning student autonomy and schedule flexibility [10]. However, research has shown that faculty perceptions on whether hybrid instruction is better than face to face instruction is varied. Approximately 41% of medical schools (43 of 106) agreed that hybrid instruction was better than face to face yet approximately 23% (24 of 106) disagreed [7]. There is also a growing body of research that revealed that the move to virtual learning has increased student stress and decreased student engagement although face to face (in person) significantly limits the number of students you can engage and the access to experts (e.g., alumni) to participate [7]. It is imperative to examine the effects of COVID-19 on pathway programs as it has a direct and indirect effect on underrepresented learning communities and therefore the diversity of future healthcare professionals [7]. Moreover, more research is needed to identify which instruction delivery is most effective for student learning. The current study will provide preliminary data to address these gaps in the literature.

### Current Study

This exploratory study examined whether in person, virtual, or hybrid learning would be rated the highest in satisfaction for underrepresented pre-health undergraduate students engaged in a six-week interprofessional health pathway program. We hypothesized that virtual instruction would be receive the lowest satisfaction for method of teaching because it significantly decreases the science socialization, network development, and educational endowments of marginalized students.

The Summer Health Professions Education Program (SHPEP) is a free summer enrichment program focused on improving access to information and resources for underrepresented college students in their first two years of study and are interested in a career in medicine, dentistry, or nursing [11]. These students include, but are not limited to, individuals who identify as African American/Black, American Indian and Alaska Native, Hispanic/Latino, and from communities of socioeconomic and educational disadvantage. SHPEP’s goal is to strengthen the academic proficiency and career development of students underrepresented in the health professions and prepare them for a successful application and matriculation to health professions schools [11].

Using a quality improvement survey, we investigated the difference in satisfaction scores for the overall program (science socialization), learning/social experiences (network development), and core curriculum courses (educational endowments) when instruction was delivered in person, virtually, and a hybrid instruction.

## 2. Methods

### 2.1. Participants

Participants were recruited from the University of Texas Health Science Center at Houston SHPEP. The university has a maximum allocation of 80 student positions. For this study, data were collected and analyzed from the years 2018 (in-person, pre-pandemic), 2020 (virtual, during pandemic), and 2022 (hybrid, post-peak pandemic). The 80 annual student positions were from the following concentrations: 40 students designated toward pre-medicine, 20 students toward pre-dentistry, and an additional 20 students toward pre-nursing. At the conclusion of the six-week program, participants were prompted to complete an online pathway program satisfaction survey.

### 2.2. Program Instruction

During the virtual and hybrid year, students were provided resources for internet services if they reported not having any prior to the program start. Comparative to in person instruction, during the virtual and hybrid year, participants were encouraged to treat this internship as a job and devote their full attention to the coursework and minimize distractions to the best of their ability. Technical support was provided through the UTHealth IT department for potential operating system issues. Across in person, virtual, and hybrid instruction, each cohort were provided a URiM mentor that facilitated student to student interaction and further educational interaction if requested.

### 2.3. Survey

At the conclusion of the six-week program, participants were required to complete a twenty-question online satisfaction survey. Of the twenty-question survey, the current study focuses on the following four, “Overall, what is your level of satisfaction with UTHealth SHPEP?”, “How satisfied were you with the learning/social experiences in SHPEP”, “How satisfied are you that the basic science core curriculum increased your knowledge in Anatomy and Physiology?”, and “How satisfied are you that the basic science core curriculum increased your knowledge in Organic Chemistry?” Overall level of satisfaction with the SHPEP program was used as a proxy for science socialization which examines the connection between scientific careers and the ability to serve community goals because this was the marketed purpose and goal of the SHPEP program. Satisfaction with learning experiences was used as a proxy measure for network development because we intentionally chose faculty members and student mentors from underrepresented backgrounds to provide the learning and social experiences. Lastly, educational endowments were measured through Anatomy, Physiology and Organic Chemistry because these courses provided study materials, small group study session, and innovative lectures. Participant responses scored on a 3-point Likert scale (1 = very satisfied to 3 = not satisfied). For each question, if a participant selected “not satisfied” they were asked to describe improvements that would better support them in that area. Across each year, courses offered, and access tomentors remained the same.

### 2.4. Analysis

Data were collected from each cohort that coincided with the three instructional formats; 2018-pre-pandemic in which the program was offered fully in person, 2020-pandemic in which the program was offered virtually, and 2022-post-peak phase in which the program was offered in a hybrid format (2 weeks virtual, 4 weeks in person). Descriptive statistics were used to analyze data across all time points. Thematic analysis was used to assess qualitative feedback. A deductive, latent approach was taken, as the researchers worked directly with the students, had knowledge of the real-time perceptions of satisfaction, and was able to place data in context within the appropriate social context.

## 3. Results

Demographics for in person, virtual, and hybrid cohorts are listed in Table 1.

52 students (65%) completed the satisfaction survey during in person instruction, 69% (n = 52) during virtual, and 90% (n = 71) during hybrid instruction.

As shown in Figure 1, science socialization satisfaction was rated highest by participants when the pathway program was delivered in hybrid instruction. Thematic analysis of the virtual instruction year revealed themes of feeling discounted. Students felt like there was valuable exposure that was not provided to them. For example, one student wrote, “it was very informational, but it definitely wasn’t the same not being able to have the in-person experience so I feel like we as a while missed out on a lot”. There were no qualitative data for in person instructional year.

Similarly, network development via learning/social experiences was also rated highest when the program was delivered in a hybrid instruction. Thematic analysis of responses during the virtual instructional year revealed an overall negative emotional state and limitation imposed on their learning. Students reported, “just wasn’t the same online”; “It was much harder to understand and feel like we learned virtually. The stethoscope, for example, it’s hard to see or know whether you are using them correctly when someone isn’t physically there.” There were no qualitative data that addressed this experience for in person instructional year.

Interestingly, for educational endowments Anatomy and Physiology was rated highest in a hybrid instruction however Organic chemistry was rated highest in satisfaction during in person instruction. We probed these findings to further understand the interaction. Similar to the results from network development, thematic analysis suggests significant limitations with virtual learning and the need for an in-person component. Specifically, “Organic chemistry was a little difficult to keep up with. I have already taken organic chemistry, but the online lectures were hard to follow due to the bouncing around of various ideas”, “The videos were rather excessive and there wasn’t any practice outside of class”.

## 4. Discussion

Overall, student satisfaction with science socialization, network development, and certain educational endowments (specifically Anatomy and Physiology) was highest during hybrid instruction and lowest during virtual instruction.

Interestingly, educational endowment via Organic Chemistry was lowest during the hybrid year. This is opposite of the pattern that was revealed for science socialization and network development. However, qualitative data revealed that Organic Chemistry classes relied heavily on virtual instruction and did not have an in-person component. Therefore, the satisfaction scores were not based on a hybrid instruction but instead virtual. This matches the pattern of the other asset bundles.

Research shows that ability to work at your own time and pace, self-directed learning, and cost effectiveness are some of the most widely cited benefits of virtual learning [12]. However, these “benefits” are not perceived similarly for IPE pathway programs that focus on underrepresented students. Pathway programs are designed for students to work together and with mentors therefore self-directed learning that is encouraged with virtual learning is a disadvantage. Moreover, virtual learning is not as cost effective for underrepresented students who are at a greater risk of not having access to a laptop or reliable internet connection [13]. Although internet services, were provided for this program, the quality of the internet service was dependent on where the student lived. For example, students who joined the program from Puerto Rico noted multiple electricity blackouts across the country that impaired their learning quality. Qualitative data collected from a multi-site medical school study in 2021 revealed, “Going virtual with some of our pathway programs required us to look at issues of equity regarding internet access and infrastructure issues folks face; Not everyone had the same type of computers, so we have some students who are working strictly from iPads. We have some students had to work from their phone” [7] This poses an additional challenge for students with limited financial resources. During the virtual year, many of our students were engaging from their homes while the rest of their family was also home. This may have provide distractions as some students share their room, or needed to use their laptop in a shared space such as a kitchen or living room. While students did their best to minimize distraction, administrators do not have the control to standardize the learning environment for all. Further developing low-income and UriM students for and recruiting them into health careers requires health center leaders to pay close attention to this intersectional aspect of their identity and experience [5].

The current study supports past research on the disadvantages of virtual instruction. Prior to the pandemic, research showed that in-person learning increased perceptions of feeling connected to the school which created a buffering effect against negative mental health symptomology, such as depression and anxiety [14]. However, researchers found that when comparing health outcomes of students attending school virtually, hybrid, or in person, students attending virtual instruction were more likely to consider suicide (13.5%, 8.4%, and 3.8%, respectively); and persistent symptoms of depression (19.1%, 15.3%, and 7.6%, respectively) [14]. Results also revealed that, virtual instruction was more prevalent among black (68.2%) and Hispanic students (69.0%) compared to white students (48.1%) [14]. Collectively, the current and past studies suggest that virtual instruction is not only least preferred for IPE learning but may also be detrimental toward student mental health.

Past research shows that students reported a preference for recorded live lectures and prerecorded lectures with live follow-up sessions as a mode of teaching in comparison to nonrecorded live lectures [15]. This research supports the current findings and suggests that hybrid learning may be the preferred method of instruction compared to in person and virtual modalities. Hybrid instruction maximizes benefits and minimizes harm [16]. Specifically, through hybrid instruction it leverages the advantages of convenience, increased interaction and learning, flexibility; reduced seat time; and decreased costs for the host institution. Hybrid also observes health and safety guidelines during the post-peak phase of the COVID-19 pandemic [16]. The main disadvantage of the fully in-person model is lack of accessibility, especially for those who cannot physically travel to campus [16]. On the other hand, the main disadvantages of the fully remote model are (1) lack of human connection, and (2) lack of access to all campus resources [16]. Combining in-person and remote methodologies while observing health and safety guidelines allows for a more accessible and flexible program that also promotes deeper human connection and provides access to all instituional resources which is paramount for education endowments of the asset bundle model.

Research that examined blended, online, and in-person academic IPE learning found the greatest predictor of learner retention was time spent engaged [17]. This suggests that intentional interaction that fosters perceptions of connection and trust may increase the social networks and the cognitive engagement builds group cohesion and a science identity to which UriM students belong too [17].

Because there is still a dearth of research on the satisfaction of hybrid learning, this research may contribute to the growing literature on learning methodologies for healthcare trainees. Inclusion of hybrid courses in academic programs could lead to improvement in healthcare diversity. It is important to understand the most effective instructional format because we do not want to perpetuate the educational inequities many of us are trying to mitigate.

### Limitations and Future Resarch

It should be noted that this research is a preliminary study that used a cross sectional design based on satisfaction survey of 80 students. Although the satisfaction questionnaire is based on a three-point Likert scale, qualitative data also provides rich information in which to contextualize the Likert responses received. A possible explanation as to why virtual learning was rated so low is because the pandemic brought additional stress outside of learning. Information overload, rumors and misinformation can createan atmosphere in which one may feel overwhlemed. Moreover, worries about health, pressures related to going to work, parents working from home or the potential of job loss and consequently income loss, and declining family health were all cited as contributors to stress during the onset of the COVID-19 pandemic [18]. Future research should examine these three instructions during this high vaccination period (post-peak phase) where the woes of the pandemic are not so new. Another limitation of the current study is that we did not measure the mental health symptomology of students across the three time periods. Therefore, we could not replicate past findings that correlate instruction methodology and mental health status of students. However, qualitative data did provide further evidence that indicated a negative emotional state specifically during the virtual instructional year. While all instructors across the three instructional modalities were encouraged to use active learning pedagogy, individual teaching strategies and quality of materials may also be a predictor of differing levels of appreciation between instructional years. Although, the majority of instructors overlapped between each year, future research should examine these variables. Moreover, future research should also continue to examine instruction modality for pathway/pipeline programs including those that are shorter in length or designed differently. In the current study, t hybrid instructional year consisted of 2 weeks virtual followed by 4 weeks in person. Would the results be the same if 3 weeks were virtual and 3 weeks were in person? Or if 4 weeks were virtual and 2 weeks were in person?

## 5. Conclusions

In conclusion, mode of instruction was associated with satisfaction scores within a six-week IPE pre-health pathway program. Overall, pre-health pathway programs are preferred in a hybrid format and least preferred when presented in virtual instruction. Qualitative data suggests that virtual instruction increases feelings of isolation and at times complicates educational information due to the limitations of virtual streaming. Nonetheless, the advantages of virtual instruction are leveraged when paired with in person instruction thus creating a hybrid experience for active learning. This design can arguably improve the science socialization, educational endowments, and network development of underrepresented students within pathway programs.

## Figures and Tables

**Figure 1 healthcare-10-02399-f001:**
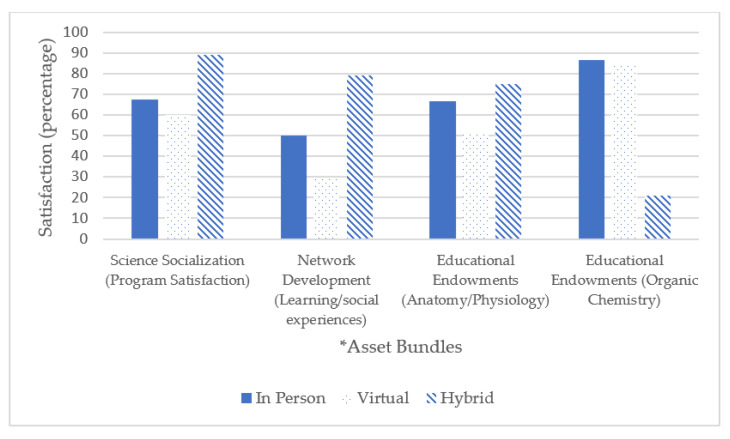
Satisfaction scores for asset bundles based on delivery format. * This study focuses on three asset bundle domains; (1) science socialization-the connection between scientific careers and the ability to serve community goals, (2) network development-building positive social capital through mentoring relationships and involvement in extracurricular activities, students, and (3) educational endowments-access to additional resources and educational support including additional study material and innovative lectures.

**Table 1 healthcare-10-02399-t001:** Participant demographics.

	In Person Pre-Pandemic 2018 N (%)	Virtual Pandemic 2020 N (%)	Hybrid Post Peak Phase 2022 N (%)
Black	40 (50%)	57 (76%)	44 (56%)
White	3 (4.0%)	2 (2.0%)	8 (10%)
Asian	14 (18%)	7 (9.0%)	12 (15%)
Native American/Pacific Islander	2 (3.0%)	3 (4.0%)	2 (2.0%)
Multi/Other	2 (3.0%)	8 (10%)	13 (17%)
Ethnicity: Hispanic	26 (33%)	29 (39%)	26 (33%)
Sex (Female)	57 (71%)	52 (69%)	63 (80%)
Age (Avg)	20	20	20
Reduced Lunch *	N/A	39 (52%)	33 (42%)
Pell Grant Recipient *	N/A	48 (64%)	36 (45%)
Total	80	75	79

* Although Reduced Lunch and Pell Grant Recipient data was not collected during the in-person year, 65% did indicate that they came from a disadvantaged background.

## Data Availability

Remaining data are available from the corresponding author upon reasonable request.

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
