# Peer review of "In-Person, Remote, or Hybrid Instruction? A Quality Improvement Assessment of a Six Week Interprofessional Education Pathway Program for Undergraduate Pre-Health Students"

_healthcare, 2022, doi:10.3390/healthcare10122399_

Round 1

Reviewer 1 Report

Significant modifications to the theoretical content of the article are recommended.
In the introduction and theoretical framework, the difficulties of access of ethnic minorities to health care and the need to promote diversity and multiculturalism in health care as a basic objective are mentioned. It seems that the article is related to these contents, which are not really worked out empirically. The article focuses on analysing the differences (advantages and disadvantages) of hybrid and face-to-face virtual teaching models from a given experience in a group of students who happen to be ethnically and racially diverse. But they do not work
In this sense, it is recommended to provide internal coherence to the study, either the specific differences and potential difficulties in access and development of teaching to the groups are explained on the basis of the multiculturalism alluded to, or this section is simply eliminated (replacing it with a theoretical framework on types of non-virtual virtual teaching) and only discusses such ethnic and racial differences when the summer course is contextualised.
Not doing so contradicts the essence of multiculturalism and anti-discrimination.

There is room in the theoretical framework for a further explanation of the difficulties of access to health care bodies in the USA for ethnic minorities; where does the problem lie? We understand that it is due to economic differences, validation of qualifications... and it is not made clear. Failure to explain this can lead to prejudices that support a false differentiation in terms of skills and or qualifications, which encourages possible bias and discrimination while killing the objectives of multiculturalism.

It alludes to difficulties and advantages of implementing distance education, but does not refer to any of them; which I understand is directly related to (mediates, directly influences) the promotion of diversity that the article discusses. The coexistence, the direct relationship are key elements in the generation of beliefs of equality, in the elimination of the perception of the different, for example. Key elements that eliminate distance.
The fact of dealing with groups with access difficulties necessarily implies verifying whether they have access to the programme with equal needs in terms of resources, technical means, etc. (material conditions, time, etc.), something that is not made clear in the study either.
In conclusion, allusion is made to diversity and multiculturalism, but the study does not analyse any of this (from line 71 to 83), it is based on analysing which model of face-to-face, virtual or hybrid learning is more effective for undergraduate health students.

Benhabib, S., 2002, The Claims of Culture: Equality and Diversity in the Global Era , Princeton: Princeton University Press.

Eisenberg, A. y J. Spinner-Halev (eds.), 2005, Minorías dentro de minorías: igualdad, derechos y diversidad , Cambridge: Cambridge University Press.

Gitlin, T., 1995, The Twilight of Common Dreams: Why America is Wracked by Culture Wars , Nueva York: Metropolitan Books.

Hero, R. y R. Preuhs, 2006, “Multiculturalism and Welfare Policies in the USA: A State-Level Comparative Analysis,” en Multiculturalism and the Welfare State: Recognition and Redistribution in Contemporary Democracies , K. Banting and W. Kymlicka ( eds.), Oxford: Oxford University Press, págs. 121–151.

The sample selection, procedure and methodology are clear. As are the conclusions,
It is striking that the discussion refers to theoretical elements contrasted with empirical experience on the advantages and disadvantages of one model or another, but this has not been worked out in the theoretical framework. This again shows an internal inconsistency in the article; the theoretical framework does not correspond to the empirical framework.
>From highlighting the value of the face-to-face relationship in educational processes (linked to eliminating prejudices and therefore promoting diversity) without disregarding the proven quality and value of other appropriate pedagogical designs based on technology to when the necessary resources and trained teaching staff are available to teach them (two elements not mentioned here).

Other drawbacks in distance systems (1) do all learners have appropriate connectivity and devices? (network quality, bandwidth, appropriate equipment...); (2) conditions for receiving teaching (rest, simultaneous work and classes, quantity and duration of the course, timetables and time differences...); (3) what support systems are offered during the course (tutorials, access to teaching staff outside teaching hours to follow the course...); (4) technical support to solve eventualities in the operating systems? (4) evaluation models based on the type of activities; (4) strategies that encourage personal and educational interaction between students and between each student and teacher during the sessions?

These elements and others are the ones that mark the quality of remote systems, and they are the consolidated ones in face-to-face education. It is necessary to compare systems with the prior guarantee of knowing that we are comparing the same ones, to say that some things are better or worse, work more or less, generate more or less satisfaction, we must verify that we are comparing the same ones. This is not clear to me here. I would appreciate clarification in this regard

Fox, K., Bryant, G., Lin, N., y Srinivasa, N. (2020). Time for Class – COVID-19 Edition Part 1: A National Survey of Faculty during COVID-19. Tyton Partners and Every Learner Everywhere, July 8, 32 pp. https://www.every learnereverywhere.org/resources/time-for-class-covid-19-edition

Hart, J. (2020). Top 200 Tools for Learning 2020, https://www.toptools4learning.com/ Hodges, C., Moore, S., Lockee, B., Trust, T., y Bond, A. (2020). La diferencia entre la enseñanza remota de emergencia y el aprendizaje en línea. Educause Review. https://er.educause.edu/articles/2020/3/the-difference-between-em ergency-remote-teaching-and-online-learning

 IESALC-UNESCO (2020). El coronavirus-19 y la educación superior: impacto y recomendaciones. https://cutt.ly/xd HJuhK

Jandrić P., Knox J., Besley T., Ryberg T., Suoranta J., y Hayes S. (2018). Postdigital science and education. Educational Philosophy and eory, 50(10): 893–899. https://doi.org/10.1080/00131857.2018.1454000

Johnson, N., Veletsianos, G., y Seaman, J. (2020). U.S. Faculty and Administrators’ Experiences and Approaches in the Early Weeks of the COVID-19 Pandemic. Online Learning, 24(2). https://doi.org/10.24059/olj.v24i2.2285

Y todo ello sin entrar a trabajar otros elementos claros vinculados a lo socioemocional o dimensiones psicosociales propias de la educación.

Más bibliografía sobre beneficios y dificultades de implantación de docencia a distancia y cómo ejecutarla en salud

FKetcherside W, Olson J, Hunt L, Mehta J, Pallares Gutiérrez C, Coy L, et al. Feasibility and acceptance of a cloud-based mobile app for AMS and IC. Int J Infect Control 2020; 6(3): 2–6. doi: 10.3396/ijic.v16i3.021.20 7. United Nations Educational, Scientific and Cultural

Organization (UNESCO) International Institute for Higher Education in Latin America and the Caribbean (IESALC). COVID-19 and higher education: today and tomorrow. Impact analysis, policy responses and recommendations. Paris, France: UNESCO; 2020. Available from: https://www.guninetwork.org/ files/covid-19_en_090420.pdf [cited 10 April 2020].

Jackson D, Bradbury-Jones C, Baptiste D, Gelling L, Morin K, Neville S, et al. Life in the pandemic: some reflections on nursing in the context of COVID-19. J Clin Nurs 2020; 29(13–14): 2041– 3. doi: 10.1111/jocn.15257

Author Response

Reviewer: Significant modifications to the theoretical content of the article are recommended. In the introduction and theoretical framework, the difficulties of access of ethnic minorities to health care and the need to promote diversity and multiculturalism in health care as a basic objective are mentioned.
In this sense, it is recommended to provide internal coherence to the study, either the specific differences and potential difficulties in access and development of teaching to the groups are explained on the basis of the multiculturalism alluded to, or this section is simply eliminated (replacing it with a theoretical framework on types of non-virtual virtual teaching) and only discusses such ethnic and racial differences when the summer course is contextualised.
Not doing so contradicts the essence of multiculturalism and anti-discrimination.

Answer: Thank you for this. This concern was not brought up by the other two reviewers. The summer course in contextualized as it is focused on underrepresented students and thus we discuss pathway programs as a whole and the disparities in healthcare workers. Moreover, we delve into the barriers of access of ethnic minorities to health care and the need to promote diversity by stating,  Historically, lack of financial resources, sole focus on standardized test scores vs holistic admission, lack of URiM identified faculty, and lack of URiM role models have been cited as barriers to racial/ethnic minorities entering health careers. Therefore, pathway programs were created to mitigate these barriers and increase the number of racial/ethnic minority students who apply to a health career ". We then discuss how the current event of COVID-19 , has prompted practioners to understand how the varied instructional methods effects of satisfaction of the program. In discussing multicultralism, we state, "Moreover, virtual learning is not as cost effective for underrepresented students who are at a great risk of not having access to a laptop or reliable internet connection [15]. Although internet services, were provided for this program, the quality of the internet service was dependent on where the student lived. For example, students who joined the program from Puerto Rico noted multiple electricity blackouts across the country that impaired their learning quality. Qualitative data collected from a multi-site medical school study in 2021 revealed, “Going virtual with some of our pathway programs required us to look at issues of equity regarding internet access and infrastructure issues folks face; Not everyone had the same type of computers, so we have some students who are working strictly from iPads. We have some students had to work from their phone” [7] This poses an additional challenge for students with limited financial resources. Further developing low-income and UriM students for and recruiting them into health careers requires health center leaders to pay close attention to this intersectional aspect of their identity and experience [5]. "

Reviewer: There is room in the theoretical framework for a further explanation of the difficulties of access to health care bodies in the USA for ethnic minorities; where does the problem lie? We understand that it is due to economic differences, validation of qualifications... and it is not made clear. Failure to explain this can lead to prejudices that support a false differentiation in terms of skills and or qualifications, which encourages possible bias and discrimination while killing the objectives of multiculturalism.

Answer: Line 44-49 now state, “Historically, financial resources, sole focus on standardized test scores vs holistic admission, lack of URiM identified faculty, and lack of URiM role models have been cited as barriers to racial/ethnic minorities who are interested in pursuing a health career. Therefore, pathway programs were created to mitigate these barriers and increase the number of racial/ethnic minority students who apply to a health career.”

Reviewer: It alludes to difficulties and advantages of implementing distance education, but does not refer to any of them; which I understand is directly related to (mediates, directly influences) the promotion of diversity that the article discusses. The coexistence, the direct relationship are key elements in the generation of beliefs of equality, in the elimination of the perception of the different, for example. Key elements that eliminate distance.

Answer: The reviewer states that we do not refer to the difficulties and advanatages to implementing distance education but we discuss the pros and cons of the format in in the introduction and in the discussion. Specifically we state in lines 114-131,

“It is difficult to differentiate the efficacy of instruction delivery formats. Emerging research shows that there are various advantages among delivery formats [8]. Face to face instruction (also known as “in person”) can provide deeper understanding through teacher and other students' body language and voice, while virtual instruction encourages student self-directed learning and to take on more responsibility for their own acquisition of knowledge, while hybrid learning student autonomy and schedule flexibility [8]. However, research has shown that faculty perceptions on whether hybrid/ blended instruction is better than face to face instruction is varied. Approximately 41% of medical schools (43 of 106) agreed that hybrid instruction was better than face to face yet approximately 23% (24 of 106) disagreed [5]. There is also a growing body of research that revealed that the move to virtual learning has increased student stress and decreased student engagement although face to face (in person) significantly limits the number of students you can engage and the access to experts (e.g., alumni) to participate [5]. It is imperative to examine the effects of COVID-19 on pathway programs as it has a direct and indirect effect on underrepresented learning communities and therefore the diversity of future healthcare professionals [5]. Moreover, more research is needed to identify which instruction delivery is most effective for student learning. The current study will address these gaps in the literature.

We further explore the relationship in the discussion in lines 251- 289. For space purposes we did not copy and paste.

Reviewer: The fact of dealing with groups with access difficulties necessarily implies verifying whether they have access to the programme with equal needs in terms of resources, technical means, etc. (material conditions, time, etc.), something that is not made clear in the study either.
In conclusion, allusion is made to diversity and multiculturalism, but the study does not analyse any of this (from line 71 to 83), it is based on analysing which model of face-to-face, virtual or hybrid learning is more effective for undergraduate health students.

Answer: Line 160-161 now reads, “During the virtual and hybrid year, students were provided resources for internet services if they reported not having any prior to the program start.” Furthermore, we analyze elements of diversity and multiculralism in the discussion stating, " Although internet services, were provided for this program, the quality of the internet service was dependent on where the student lived. For example, students who joined the program from Puerto Rico noted multiple electricity blackouts across the country that impaired their learning quality. Qualitative data collected from a multi-site medical school study in 2021 revealed, “Going virtual with some of our pathway programs required us to look at issues of equity regarding internet access and infrastructure issues folks face; Not everyone had the same type of computers, so we have some students who are working strictly from iPads. We have some students had to work from their phone” [5] This poses an additional challenge for students with limited financial resources. Further developing low-income and UriM students for and recruiting them into health careers requires health center leaders to pay close attention to this intersectional aspect of their identity and experience [4].

Reviewer: The sample selection, procedure and methodology are clear. As are the conclusions, It is striking that the discussion refers to theoretical elements contrasted with empirical experience on the advantages and disadvantages of one model or another, but this has not been worked out in the theoretical framework. This again shows an internal inconsistency in the article; the theoretical framework does not correspond to the empirical framework.
>From highlighting the value of the face-to-face relationship in educational processes (linked to eliminating prejudices and therefore promoting diversity) without disregarding the proven quality and value of other appropriate pedagogical designs based on technology to when the necessary resources and trained teaching staff are available to teach them (two elements not mentioned here).

Answer: We discuss how the program is built with URM faculty members and mentors which has been shown to reduce prejudice and promote diversity. We also state how each cohort is provided a mentor and UTHealth IT who have the resources and are trained in teaching. This is now a section in Methods

Reviewer: Other drawbacks in distance systems (1) do all learners have appropriate connectivity and devices? (network quality, bandwidth, appropriate equipment...); (2) conditions for receiving teaching (rest, simultaneous work and classes, quantity and duration of the course, timetables and time differences...); (3) what support systems are offered during the course (tutorials, access to teaching staff outside teaching hours to follow the course...); (4) technical support to solve eventualities in the operating systems? (4) evaluation models based on the type of activities; (4) strategies that encourage personal and educational interaction between students and between each student and teacher during the sessions?
These elements and others are the ones that mark the quality of remote systems, and they are the consolidated ones in face-to-face education. It is necessary to compare systems with the prior guarantee of knowing that we are comparing the same ones, to say that some things are better or worse, work more or less, generate more or less satisfaction, we must verify that we are comparing the same ones. This is not clear to me here. I would appreciate clarification in this regard

Answer: This is a great point. The methods now include a section labeled Program instruction. It now reads:

Program Instruction

During the virtual and hybrid year, students were provided resources for internet services if they reported not having any prior to the program start. Comparative to in person instruction, during the virtual and hybrid year, participants were encouraged to treat this internship as a job and devote their full attention to the coursework and minimize distractions to the best of their ability. Technical support was provided through the UTHealth IT department for potential operating system issues. Across in person, virtual, and hybrid instruction, each cohort were provided a mentor that facilitated student to student interaction and further educational interaction if requested.

We also discuss in the discussion:

During the virtual year, many of our students were engaging from their homes while the rest of their family was also home. This may have provide distractions as some students share their room, or needed to use the laptop in a shared space such as a kitchen or living room. While students did their best to minimize distraction, administrators do not have the control to standardize the learning environment for all. 

Reviewer 2 Report

1.      This study lacks innovation, and the authors should compare similar literature to highlight differences and innovations.

2.      The research lacks data and contents.

Author Response

Reviewer:  This study lacks innovation, and the authors should compare similar literature to highlight differences and innovations.

Answer: To our knowledge, this is the first study to investigate whether mode of instruction affects the science socialization, network development, and educational endowments of underrepresented college students in a pathway program geared toward health career. With the increase in health disparities that COVID-19 has helped illuminate, the need for this research is paramount as many health centers are creating pathway programs from scratch. While the implications are straightforward, the innovative approach of using theoretical asset bundles and measuring its affect via three modes of instruction I think are imperative and have implications for many diversity administrators.

To compare similar literature and highlight differences lines 83-91 reads, “In a study that examined a cascading mentorship model for a medical pathway program, researchers found that having by faculty members from underrepresented backgrounds and mentors who were close to the age of the students offset potential deficits in the “asset bundles” of science socialization and network development because they were able to adequately address concerns over low expectations about academic ability, fear of antagonism from the dominant group, and low visibility of others with similar backgrounds [5] The current study will expand past research by first examining an IPE pathway program with underrepresented college students but also measuring how mode of instruction affects their satisfaction with the asset bundles.

We also make explicit the gaps in the literature in lines 115-126 that states, “[8]. However, research has shown that faculty perceptions on whether hybrid/ blended instruction is better than face to face instruction is varied. Approximately 41% of medical schools (43 of 106) agreed that hybrid instruction was better than face to face yet approximately 23% (24 of 106) disagreed [5]. There is also a growing body of research that revealed that the move to virtual learning has increased student stress and decreased student engagement although face to face (in person) significantly limits the number of students you can engage and the access to experts (e.g., alumni) to participate [5]. It is imperative to examine the effects of COVID-19 on pathway programs as it has a direct and indirect effect on underrepresented learning communities and therefore the diversity of future healthcare professionals [5]. Moreover, more research is needed to identify which instruction delivery is most effective for student learning. The current study will address these gaps in the literature.

Reviewer: The research lacks data and contents.

Answer: This study provides quantitative data from three time points and qualitative data from three time points. We address the limitations and the need for future research in our discussion section as this research should serve as a preliminary study.

Reviewer 3 Report

This paper deals with the evaluation of the delivery modality (in-person, remote or hybrid) of a 6-week pathway program for racial/ethnic minorities in their first two years of study interested in a career in medicine, dentistry, or nursing. 

The issues identified in this paper (emerging learning modalities; equity in education) are current problems in education, and there is a need for studies focusing on them. 

This paper tackles these themes through the analysis of 4 questions included in a satisfaction survey administered at the end of 3 editions of the program, each one delivered in a different modality. 

The paper is clear and well-written, and it is rich in interesting prompts. However, there are a few aspects that could be improved before acceptance.

1) The introduction should be enriched with more current studies on digital education and equity in education. For example: lines 44-48: the authors identify 3 aspects on which to focus the analysis of the questionnaire. Why are they important? Are there other studies focusing on these aspects? Moreover, in the discussion (lines 157-164) studies on the efficacy of the different learning modalities are mentioned. These and other studies should be described more extensively in the introduction, and just mentioned in the discussion. 

2) The methodology is rather weak since the sample is small, the analysis relies on few items of a satisfaction questionnaire, and a 3-points Likert scale is used. The authors should demonstrate to be aware of these limitations. 

3) The authors should explicitly explain why the 4 questions object of analysis are connected to the three aspects on which they focus the analysis, and justify why they chose these items. In my opinion, these questions are quite vague. 

4) The method section should be enriched with details explaining if the three editions of the course differ in other details, if the teachers were the same in the 3 editions, which learning approaches were used. The way a course is organized and the learning approaches used could affect the results and satisfaction more than the modality (in-person, remote, or hybrid). It does not seem that the 4 questions can capture this difference. That is why I am worried about the weakness of this paper. 

Author Response

Reviewer: The introduction should be enriched with more current studies on digital education and equity in education. For example: lines 44-48: the authors identify 3 aspects on which to focus the analysis of the questionnaire. Why are they important? Are there other studies focusing on these aspects? Moreover, in the discussion (lines 157-164) studies on the efficacy of the different learning modalities are mentioned. These and other studies should be described more extensively in the introduction, and just mentioned in the discussion. 

Answer: Thank you for this feedback. Lines 43- 49 read, “One type of pathway program that has gained popularity over the past few decades due to the collaborative nature of the healthcare environment is interprofessional education (IPE) pathway programs. IPE pathway programs provides students with the opportunity to learn about, from, and with multiple healthcare profession in order to develop interprofessional collaborative working relationships that enable the highest quality of care across settings and therefore optimal health outcomes [2,3]. “ Therefore, I am unsure what the author is referring too.

However, lines 53-58 read, “Asset bundles are defined as the specific sets of abilities and resources individuals need to succeed in educational and professional tasks [4]. Collectively, there are five asset bundles; educational endowments, science socialization, network development, family expectations, and material resources [4].  While asset bundles may affect the educational achievement of students from any background, researchers Johnson and Bozeman purport that understanding he interaction of these variables for minorities belonging to multiple stigmatized identity groups is central to the advancement of diversity in healthcare.” This may be what the reviewer is referring too. If yes, I describe that these assets are important to address because they are believed to decrease UriM attrition in the health science pipeline. This stated in lines 52-53 and further explicated in lines 61-67.

As the reviewer suggested, I added another study that examined the asset bundles in a pathway program. Lines 83-89 now reads, “In a study that examined a cascading mentorship model for a medical pathway program, researchers found that having by faculty members from underrepresented backgrounds and mentors who were close to the age of the students offset potential deficits in the “asset bundles” of science socialization and network development because they were able to adequately address concerns over low expectations about academic ability, fear of antagonism from the dominant group, and low visibility of others with similar backgrounds [5]”

To expound the efficacy of the different learning modalities in the introduction lines 106-122 read, “It is difficult to differentiate the efficacy of instruction delivery formats. Emerging research shows that there are various advantages among delivery formats [8]. Face to face instruction (also known as “in person”) can provide deeper understanding through teacher and other students' body language and voice, while virtual instruction encourages student self-directed learning and to take on more responsibility for their own acquisition of knowledge, while hybrid learning student autonomy and schedule flexibility [8]. However, research has shown that faculty perceptions on whether hybrid/ blended instruction is better than face to face instruction is varied. Approximately 41% of medical schools (43 of 106) agreed that hybrid instruction was better than face to face yet approximately 23% (24 of 106) disagreed [5]. There is also a growing body of research that revealed that the move to virtual learning has increased student stress and decreased student engagement although face to face (in person) significantly limits the number of students you can engage and the access to experts (e.g., alumni) to participate”

Reviewer: The methodology is rather weak since the sample is small, the analysis relies on few items of a satisfaction questionnaire, and a 3-points Likert scale is used. The authors should demonstrate to be aware of these limitations. 

Answer:  To be more specific with the methodology, we have added an analysis section that reads, “Analysis- Data were collected from each cohort that coincided with the three instructional formats; 2018 -pre-pandemic in which the program was offered fully in person, 2020- pandemic in which the program was offered virtually, and 2022 -post-peak phase in which the program was offered in a hybrid format (2 weeks virtual, 4 weeks in person). Descriptive statistics were used to analyze data across all time points. Thematic analysis was used to assess qualitative feedback. A deductive, latent approach was taken, as the researchers worked directly with the students, had knowledge of the real-time perceptions of satisfaction, and was able to place data in context within the appropriate social context.

We also added in the limitation section that in lines 292- 295, “It should be noted that this research is a cross sectional design based on satisfaction survey of 80 students. Although the satisfaction questionnaire is based on a 3-point Likert scale, qualitative data also provides information in which to contextualize the Likert responses received.

Reviewer: The authors should explicitly explain why the 4 questions object of analysis are connected to the three aspects on which they focus the analysis, and justify why they chose these items. In my opinion, these questions are quite vague. 

Answer: Under the survey section from lines 164-173, it now states, “Overall level of satisfaction with the SHPEP program was used as a proxy for science socialization which examines the connection between scientific careers and the ability to serve community goals because this was the marketed purpose and goal of the SHPEP program. Satisfaction with learning experiences was used as a proxy measure for network development because we intentionally chose faculty members and student mentors from underrepresented backgrounds to provide the learning and social experiences. Lastly, educational endowments were measured through Anatomy, Physiology and Organic Chemistry because these courses provided study materials, small group study session, and innovative lectures”.

Reviewer: The method section should be enriched with details explaining if the three editions of the course differ in other details, if the teachers were the same in the 3 editions, which learning approaches were used. The way a course is organized and the learning approaches used could affect the results and satisfaction more than the modality (in-person, remote, or hybrid). It does not seem that the 4 questions can capture this difference. That is why I am worried about the weakness of this paper. 

Answer: We added a "Program Instruction" section that states, "During the virtual and hybrid year, students were provided resources for internet services if they reported not having any prior to the program start. Comparative to in person instruction, during the virtual and hybrid year, participants were encouraged to treat this internship as a job and devote their full attention to the coursework and minimize distractions to the best of their ability. Technical support was provided through the UTHealth IT department for potential operating system issues. Across in person, virtual, and hybrid instruction, each cohort were provided a mentor that facilitated student to student interaction and further educational interaction if requested.

Also added the following lines to the method section," Across each year, courses offered, course instructor, and number of mentors remained the same.”

Round 2

Reviewer 1 Report

The internal coherence of the study has improved significantly with the improvements introduced. The publication ys recommended 

  •  

Author Response

Thank you sincerely for recommending this for publication ! We are overjoyed.

Reviewer 2 Report

The paper has been revised based on the comments.

Author Response

Thank you for acknowleding our revisions are proficient. We have reviewed and used the resources provided for the fine/minor spell check.

Reviewer 3 Report

Dear authors,

thank you for considering my comments when revising the paper. It noticeably improved after the revisions. 

However, I still have some concerns. 

1) in line 112 you wrote that hybrid and blended learning are the same thing. This is not true for many authors. It would be better to add some citations to support this statement, or change it. 

2) It would be better to anticipate the model of hybrid education, mentioned in line 201, in the method section (such as in line 161). 

4) Lines 315-323 should be moved to the introduction and the references just mentioned here to compare and discuss the results of this study. In the conclusions there should not be new references. 

5) lines 327-329: the instructional mode (intended as virtual, hybrid or in-person teaching) is not the only thing that makes a difference when the issue is uneven inequalities. What really makes the difference are the teaching strategies and approaches, and the quality of the materials and activities, and they can be independent of the instructional mode. I suggest that authors add a reflection in this direction. It would also be interesting to know what pedagogies have been used in the three modalities because maybe they were the very cause of the different levels of appreciation of the courses. 

6) I still perceive a discrepancy between what the authors intended to measure and the questionnaire's items. In particular, the questionnaire asks the students' satisfaction, and it does not measure the effectiveness of the teaching modalities. So, I suggest removing all the references to effectiveness (such as line 358). Maybe the author might add the intention to measure the effectiveness of the three different modalities among the further directions. 

Author Response

Happy Thanksgiving Reviewers !

  1. in line 112 you wrote that hybrid and blended learning are the same thing. This is not true for many authors. It would be better to add some citations to support this statement, or change it. 

Answer: All statements that refer to hybrid and blended as the same have been removed. Specifically, line 109 now read,Hybrid learning, is an approach to education that combines online educational materials with traditional in-person classroom methods. And line 117 reads, “However, research has shown that faculty perceptions on whether hybrid instruction is better than face to face instruction is varied.”

2. It would be better to anticipate the model of hybrid education, mentioned in line 201, in the method section (such as in line 161). 

Answer: I am unsure what the reviewer is referring too. Line 161 in the methods sections reads,Comparative to in person instruction, during the virtual and hybrid year, participants were encouraged to treat this internship as a job and devote their full attention to the coursework and minimize distractions to the best of their ability.” Line 201 is a footnote and the following reads, “52 students (65%) completed the satisfaction survey during in person instruction, 69% (n=52) during virtual, and 90% (n=71) during hybrid instruction. “

The hypothesis is listed under the "Current Study" section on line 134 that states, "We hypothesized that virtual instruction would be receive the lowest satisfaction for method of teaching because it significantly decreases the science socialization, network development, and educational endowments of marginalized students. " Is this what the reviewer meant by anticipating hybrid learning?

4) Lines 315-323 should be moved to the introduction and the references just mentioned here to compare and discuss the results of this study. In the conclusions there should not be new references. 

Answer: Lines 315- 323, states, "Psychological safety is the term used to describe a learning environment where people feel comfortable speaking up to share without fear of dismissal, rejection, embarrassment, repercussion, and ridicule [20]. This concept may be central to all asset bundle examined including educational endowment, science socialization, and network development.

Because there is still a dearth of research on the effectiveness of hybrid learning, this research may contribute to the growing literature on learning methodologies for healthcare trainees. Inclusion of hybrid courses in academic programs could lead to improvement in healthcare diversity."

The first paragraph and other mentions of psychological safety are now removed from the entirety of the manuscript.

5) lines 327-329: the instructional mode (intended as virtual, hybrid or in-person teaching) is not the only thing that makes a difference when the issue is uneven inequalities. What really makes the difference are the teaching strategies and approaches, and the quality of the materials and activities, and they can be independent of the instructional mode. I suggest that authors add a reflection in this direction. It would also be interesting to know what pedagogies have been used in the three modalities because maybe they were the very cause of the different levels of appreciation of the courses. 

Answer:   in lines 329-333, as suggested, we have now added, “While all instructors across the three instructional modalities were encouraged to use active learning pedagogy, individual teaching strategies and quality of materials may also be a predictor of differing levels of appreciation between instructional years. Although, the majority of instructors overlapped between each year, future research should examine these variables. "

6) I still perceive a discrepancy between what the authors intended to measure and the questionnaire's items. In particular, the questionnaire asks the students' satisfaction, and it does not measure the effectiveness of the teaching modalities. So, I suggest removing all the references to effectiveness (such as line 358). Maybe the author might add the intention to measure the effectiveness of the three different modalities among the further directions. 

Answer: Throughout the entirety of the paper, all references of effectiveness were replaced with rating of satisfaction and preference.